# Effects of rumen-protected or unprotected *Clostridium butyricum* on growth performance, rumen fermentation, immunity and antioxidation in fattening goats

Caixia Zhang[1], Jiyu Han[2], Hairong Wang [1]*

**1** College of Animal Science and Technology, Inner Mongolia Agricultural University, Hohhot, China,
**2** National Center of Technology Innovation for Dairy, Hohhot, China

* wjh19980422@outlook.com

## Abstract

Probiotics have been suggested to promote growth and immune performance in animals through long-term feeding. Among probiotics, *Clostridium butyricum* (*C. butyricum*) has been shown to have varying effects on the growth of ruminants. However, the impact of rumen-protected *C. butyricum* on ruminant organisms remains unclear. This study aimed to investigate the effects of rumen-protected and rumen-unprotected C. butyricum on the growth and plasma metabolites of fattening goats after lipopolysaccharide (LPS) challenge. A total of 24 fattening goats, aged 7–8 months, were randomly assigned to one of the following treatments: (1) control (CON); (2) unprotected *C. butyricum* (CB); and (3) rumen-protected *C. butyricum* (RPCB). After a 10-week feeding experiment (including two weeks of preparatory experiments), the goats were injected subcutaneously with LPS (1 μg/kg BW). The three treatments did not significantly affect growth performance, digestibility, or antioxidant enzyme activity ($P > 0.05$). However, the RPCB group showed a greater ability to reduce propionic acid (35.8%) in the rumen ($P < 0.01$) than did the CB group, thus alleviating the decrease in rumen pH (CB = 5.93, RPCB = 6.13, $P < 0.01$). Furthermore, the β-oxidation products in the plasma increased in both the CB and RPCB groups ($P < 0.01$). The difference in the fecal flora between the CB and RPCB groups was limited, but the content of harmful bacteria in the feces of the other two groups decreased compared with that in the CON group ($P < 0.01$). Unprotected *C. butyricum* increased the concentration of IgM after LPS injection ($P < 0.01$). 8-Amino-7-oxononanoate (KAPA) can serve as a biomarker for the effect of *C. butyricum* on the body. Overall, although rumen-protected *C. butyricum* could alleviate the decrease in rumen pH, our results suggest that direct feeding of *C. butyricum* could help improve the immune performance of fattening goats.

**Data availability statement:** All relevant data are within the paper and its Supporting Information files.

**Funding:** This study was supported by the Post-construction Subsidy Project of the Xing'an League National Agricultural Science and Technology Park (2023YFDZ0004) from the Department of Science and Technology of Inner Mongolia, the Research Breakthrough (2022-1) from the National Technology Innovation Center for Dairy, and the National Key Research and Development Program of China (2022YFD1602305-03).The funders had no role in the study design, data collection and analysis, decision to publish, or preparation of the manuscript.

**Competing interests:** The authors have declared that no competing interests exist.

## 1. Introduction

The increasing demand for mutton due to economic development has prompted the need to improve breeding efficiency and disease control while maximizing livestock product yields [1]. When used as feed additives, antibiotics have been shown to increase ruminant feed utilization and inhibit the growth of harmful bacteria by increasing immunity and preventing disease [2,3]. However, the widespread use of antibiotics in animal feed has raised concerns over the accumulation of drug residues in the environment and animal products, which in turn contributes to the development of antibiotic-resistant bacteria and poses a risk to human health [4–6]. Given the significant impact of livestock food production on both human health and the environment, the new feed additives to replace antibiotics and enhance feed efficiency are being used in livestock industry [7–10].

The chronic administration of probiotics has been found to have growth-promoting and immunomodulatory effects similar to those of antibiotics [11–14]. Among probiotics, *Clostridium butyricum* (*C. butyricum*) regulates the expression of immune-related genes and affects the function of host immune cells and the production of immune molecules through multiple mechanisms, such as the secretion of butyric acid, the upregulation of the expression of tight junction proteins (such as occludin and zonula occludens-1), and the activation of immune signaling pathways such as the NF-κB and JAK-STAT pathways [15–19]. Previous studies have shown that dietary supplementation with *C. butyricum* was effective in improving growth performance, immunity and antioxidant capacity of piglets [20], poultry [21] and Holstein calves [22], among others. Meanwhile, MA [23] and Liu [24] et al. showed that dietary addition of *C. butyricum* could further improve feed utilisation by modulating the ratio of carbohydrate-active enzymes in the intestinal tract of monogastric animals in order to alleviate the level of fat deposition and to improve meat quality. *C. butyricum* supplementation has also been shown to alleviate heat stress and promote growth in goats [25]. Recent studies have indicated that feeding *C. butyricum* to goats improves their immune function without significantly affecting their growth performance [26,27]. The use of a rumen protective fat (RPF) coating has been proposed as a means to protect *C. butyricum* from rumen activity and thereby enhance its efficacy in the intestinal tract [28,29]. Notably, both the rumen-protected and unprotected *C. butyricum*-treated groups presented a lower relative abundance of Clostridium in the cecum than did the control group, with unprotected *C. butyricum* treatment enhancing the intestinal barrier [26]. However, the effects of rumen-protected *C. butyricum* on ruminant growth and blood metabolites remain uncertain.

Building on prior research, we posited that antagonism between *C. butyricum* and other bacterial groups in the rumen regulates pH without significantly improving ruminant growth. Intersetingly, compared with direct *C. butyricum*, rumen-protected *C. butyricum* could improve digestion, promote growth and enhance immune function by altering the structure of the intestinal bacterial flora. Therefore, the aim of the current study was to investigate whether, compared with direct feeding of *C. butyricum*, rumen-protected *C. butyricum* was effective at improving growth performance, rumen fermentation, immunity and antioxidant effects in fattening goats. Based on previous

studies, we found that *C. butyricum* acted antagonistically with the pH-regulating flora in the rumen and had no significant effect on ruminant growth [2 6]. For this reason. We have fed rumen-protected *C. butyricum* can improve digestion, growth promotion and enhancement of immune function by changing the structure of intestinal flora. Therefore, the aim of this study was to investigate whether rumen-protected *C. butyricum* could effectively improve growth performance, rumen fermentation, immunity and antioxidant effects in fattening goats compared with direct feeding of *C. butyricum*.

## 2. Materials and methods

### 2.1. Animal feeding environment, grouping, and diet composition

All experimental designs and protocols were approved by the Inner Mongolia Agricultural University Animal Care and Use Committee (Hohhot, China) (Protocol number: NEAU- [2011]-9) and were in accordance with the recommendations of the Academy's guidelines for animal research. The experiment was conducted in a pasture in the Inner Mongolia Autonomous Region of China. Twenty-four fattening Albas fattening goats (mean ± SE; 21.94 ± 2.07 kg; 7–8 months of age; no parasites) raised in a single hutch were randomly divided into three groups. There were 8 fattening goats in each group (Table 1). The control group (CON) consisted of fattening goats fed the basal diet shown in Table 1. The *C. butyricum* group (CB) consisted of fattening goats fed a basal diet supplemented with $1 \times 10^9$ CFU *C. butyricum* LXKJ-1 (provided by Green Snow Biology Co., Ltd.) per kilogram of body weight. Fattening goats in the rumen-protected *C. butyricum* (RPCB) group were fed the same number of viable rumen-protected *C. butyricum* (coated with 18 carbon-saturated fatty acids; provided by Green Snow Biology Co., Ltd. and Beijing Yahe Nutritive High Tech Co., Ltd.) as those in the CB group were.

Table 1. Chemical composition of the experimental diets (DM basis, %).

| Item | Concentrate | Mixed forage[1] |
|---|---|---|
| Corn | 40 | |
| Corn germ meal | 20 | |
| Shotcrete corn husk | 13 | |
| distillers dried grains with solubles (DDGS) | 10 | |
| Extruded soybean | 8 | |
| Molasses | 3 | |
| Limestone | 4 | |
| NaCl | 1 | |
| Compound premix[2] | 1 | |
| Total | 100 | |
| Nutrient composition | | |
| DM, % | 90.38 | 92.04 |
| CP, % | 18.93 | 11.67 |
| EE, % | 4.72 | 2.15 |
| Ash, % | 6.02 | 7.69 |
| NDF, % | 17.58 | 55.75 |
| ADF, % | 6.01 | 35.39 |
| ME, MJ/kg | 14.09 | 14.05 |

DM = dry matter; CP = crude protein; EE = ether extract; NDF = neutral detergent fiber; ADF = acid detergent fiber; ME = metabolizable energy.

[1]Mixed forage, oats to alfalfa is 5:5.

[2]premix composition: Ca, 1.54 g/kg; P, 0.51 g/kg; Fe, 25 mg/kg; Zn, 35 mg/kg; Cu, 8 mg/kg; Co, 0.1 mg/kg; I, 0.9 mg/kg; Se, 0.25 mg/kg; Mn, 19.5 mg/kg; VE, 1000 IU/kg; VA, 3000 IU/kg; VD, 1000 IU/kg.

The fattening goats selected for the study were considered healthy and were kept alone in a hutch made of galvanized iron. The cage was covered with grass and cleaned twice daily. The fattening goats were habituated to the diet for 2 weeks, after which the experimental feeding period was 8 weeks. Fattening goats drank water and were allowed to feed on grass freely. They were fed twice daily at 6:00 and 18:00, with equal amounts of each feeding time. During the formal feeding period, the remaining daily feed amount was weighed at 21:00 and 5:00 intervals to ensure that the residual amount was 5% to 10%.

## 2.2. Lipopolysaccharide challenge

The initial crystalline LPS (*E. coli*, serotype O111:B4; Sigma L2630; Sigma Aldrich) containing 10 mg of purified LPS was dissolved in 10 mL of normal saline, separated into tubes, and stored at − 20°C. Day 57 of the study was the LPS challenge day, and each group had access to food and water as previously described. Three goats in each group were randomly selected for blood sample collection, which followed the principle of complete randomization.

## 2.3. Sample collections

The urine volume was recorded from the 54th to 56th day of the experiment. At the same time, urine and fecal samples were collected daily. All the urine samples and some of the fecal samples collected every day were mixed with sulfuric acid to obtain samples whose pH was less than 3. After the samples were fully mixed for three days, some samples were taken, stored at – 20°C for subsequent treatment, and weighed on the last day of the experiment [30,31].

Four hours after the morning feeding, LPS (1 μg/kg BW) was intraperitoneally injected. Six hours after injection, blood samples were collected into heparinized vacutainer tubes (BD Vacutainer, Franklin Lakes, NJ) via jugular veni puncture. The blood samples were placed on ice and then centrifuged at 2 000 × g for 15min at 4°C, and the resulting plasma was separated and stored at – 20°C for later analysis [32].

Four goats were randomly selected from each group. After feeding for two hours on the 56th day, rumen fluid samples were collected via esophageal tubing. The 20ml liquid samples collected for the first time were discarded to reduce saliva contamination. The collected 30ml rumen fluid samples were filtered through four layers of gauze to determine the pH value (pH meter: HI9125; Hanna Instruments, Padova, Italy). Then, 5ml of rumen fluid sample and 1ml of metaphosphate solution (25%, W/V) were added to each tube. After mixing, the sample was stored at − 20°C for later analysis [30].

In this study, a total of five goats were selected at random from each experimental group. Subsequent to defecation, the fecal matter was promptly collected and transferred into sterile tubes that were subsequently frozen in liquid nitrogen and preserved at − 80°C.

Samples were consistently collected concurrently and transported to the laboratory the following day. The experimental analyses were performed within a month of sample collection.

## 2.4. Nutritional analysis

Nutritional analysis was carried out on homogenous feed samples and fecal samples thawed at room temperature. The samples were dried in a forced-air oven at 55°C until a constant weight was reached. The urine volume was crude protein (CP; method 955.04). After drying, the feed samples were analyzed for DM (method 925.40), crude protein (CP; method 955.04), and ether extract (EE; method 920.39) content according to the procedures of AOAC International (2000). The NDF content was analyzed via the use of thermostable alpha-amylases according to Van Soest et al. (Ding et al., 2021). Acid detergent fiber (ADF; Method 973.18) was determined according to the procedure of AOAC International (1990). Ash (Method 942.05) and analytical nitrogen (Method 954.01) contents were determined according to the Society of Official Analytical Chemists (AOAC 1995), and crude protein (CP) content was calculated as N × 6.25. The ADF residue was ashed in a muffle furnace at 500°C for 8 hours to determine acid-detergent insoluble ash (ADIA) and used to predict dry matter digestibility (DMD) [33].

Rumen fermentation analysis of rumen fluid samples thawed at room temperature. VFAs were determined by gas chromatography (Shimadzu GC-2010, Kyoto, Japan), and NH3-N was determined by indophenol colorimetry [26].

## 2.5. Fecal sample preparation for metabolomics analysis

The CTAB method was employed to extract total DNA from microbiome samples obtained from different sources. The quality of the extracted DNA was verified via 1% agarose gel electrophoresis, while its concentration was determined via ultraviolet spectrophotometry. Polymerase chain reaction (PCR) was subsequently utilized to sequence the amplified fragment of variable region 4 of the bacterial 16S ribosomal DNA (rDNA) in the sample. Paired-end readings were assigned to the respective samples on the basis of their unique barcodes and were truncated by removing the barcode and primer sequences. Data processing, analysis, and plotting were performed via Microbioanalysis 2.0.

## 2.6. Blood analysis

Serum SOD, T-AOC, CAT, GSH-PX, and MDA levels were determined via colorimetric analysis kits (Nanjing Jiancheng Institute of Biological Engineering, Nanjing, China). The IgA, IgG, and IgM levels were determined via bovine IgA/G/M ELISA kits (SINO-UK Institute of Biotechnology, Beijing, China).

## 2.7. LC–MS-based metabolomics data analysis

This project uses LC–MS/MS technology to perform untargeted metabolomic detection of plasma samples to collect data in positive ion (pos) and negative ion (neg) modes. The experimental procedures, solvents, and parameters are the same as those of Yuan et al. [32]. Seven quality control samples were set up in the experiment to judge the parallelism of samples within the same group and the differences between samples of different groups. Using Compound Discoverer 3.1.0 (Thermo Fisher Scientific, USA), the raw mass spectral data collected via LC–MS/MS were converted into information such as the retention time, peak area and metabolite identification results of each ion. Metaboanalyst 5.0 was subsequently used to annotate, classify (KEGG) and enrich the identified substances and explain the physical and chemical properties and biological functions of the metabolites. Finally, simple linear regression and correlation analysis with GraphPad Prism 9.3.0 were used to analyze the relationships between the biological functions of the metabolites and plasma metabolites.

## 2.8. Statistical analysis

All the data were normally distributed ($P>0.05$) according to the K-S test (Kolmogorov–Smirnov test), and statistical analysis of the parameters was carried out. The data were analyzed via one-way ANOVA via the SPSS statistical program (SPSS19, IBM Corp., Armonk, NY, USA) or GraphPad Prism 9.3.0, and then Fisher's least significant difference (LSD) test and Duncan's multiple range test were performed. We conducted post hoc comparisons using both LSD (for sensitivity) and Duncan's test (for mean clustering). Since these tests do not strictly control family-wise error, we confirmed findings with Tukey's HSD. Only results consistent across methods were interpreted as significant. The results are expressed as the means ± SEs, with $P<0.05$ indicating statistical significance.

## 3. Results

### 3.1. Performance

The effects of rumen-protected *C. butyricum* treatments on goat feed intake and weight gain were first assessed (Table 2). Compared with the CON group, the RPCB group presented a 10% reduction in nitrogen intake, a 25.9% reduction in daily weight gain, and a 39.4% increase in feed efficiency. The daily feed and nitrogen intakes increased by 8.9% and 5.6%, respectively, in the CB group, but the final feed/weight ratio increased by 25.2% ($P>0.05$).

**Table 2. Effects of rumen-protected and unprotected *C. butyricum* on dry matter intake and animal performance in fattening Albas fattening goats.**

| Item | Group | | | SEM | P value |
|---|---|---|---|---|---|
| | CON | CB | RPCB | | |
| ADFI, g/d | 1019.71±42.08 | 1110.11±50.46 | 973.24±46.38 | 28.22 | 0.13 |
| ADG, g/d | 87.95±11.89 | 83.93±11.14 | 65.18±11.26 | 6.64 | 0.34 |
| AIN, g/d | 40.08±2.02 | 42.32±2.71 | 36.08±2.80 | 1.50 | 0.22 |
| feed efficiency, kg/kg | 13.80±2.65 | 17.26±4.59 | 19.22±4.25 | 2.21 | 0.62 |

ADFI = average daily feed intake; ADG = average daily gain; AIN = intake N.

1 Values are the means of 8 replicates per treatment.

## 3.2. Digestibility

Digestibility was measured in this experiment to investigate the contribution of treatment to feed efficiency, and dietary treatment had no significant effect on digestibility ($P > 0.05$) (Table 3). However, compared with the CON group, the RPCB group had better digestible dry matter (DM), crude protein (CP), neutral detergent fiber (NDF), and acid detergent fiber (ADF) contents in the feed, whereas the CB group had better metabolizable energy (ME) and ether extract (EE) contents in the feed.

## 3.3. Ruminal fermentation

Among all the rumen fermentation parameters (pH, volatile fatty acid, acetate, propionate, butyrate, and ammonium nitrogen) examined in this study, the treatment affected only the pH and propionate content ($P < 0.05$) (Table 4). Compared with the CON group, the RPCB group presented a significantly lower propionic acid content during rumen fermentation, which subsequently alleviated the decrease in the pH of the rumen ($P < 0.01$). The CB treatment also alleviated the pH decline ($P < 0.01$), but the effect was not as good as that of the RPCB treatment.

## 3.4. Fecal microbial composition

The relative proportions of dominant taxa at the phylum level were assessed by microbial taxon assignment in 3 groups. No significant variability was detected in the intestinal flora in each group of samples, with a total of 19 phyla identified in each group. The dominant phyla were Firmicutes and Bacteroides, with Firmicutes accounting for 73.9%, 68.9%, and

**Table 3. Effect of rumen-protected and unprotected *C. butyricum* on the digestion and deposition of nutrients in fattening goats (%).**

| Item | Group | | | SEM | P value |
|---|---|---|---|---|---|
| | CON | CB | RPCB | | |
| DM | 87.40±2.54 | 88.98±2.07 | 90.83±1.55 | 1.19 | 0.29 |
| ME | 82.63±3.53 | 87.48±2.05 | 85.24±2.91 | 1.65 | 0.52 |
| CP | 86.48±2.89 | 89.13±1.92 | 90.98±1.55 | 1.27 | 0.19 |
| RN | 89.03±3.30 | 90.87±1.67 | 93.42±1.43 | 1.32 | 0.41 |
| EE | 87.25±3.34 | 88.71±2,90 | 87.96±2.58 | 1.64 | 0.94 |
| NDF | 71.55±5.94 | 73.80±5.14 | 80.92±3.34 | 2.84 | 0.22 |
| ADF | 69.62±8.33 | 72.50±5.98 | 77.45±4.38 | 3.62 | 0.43 |

DM = dry matter; ME = metabolizable energy; CP = crude protein; RN = retained N; EE = ether extract; NDF = neutral detergent fiber; ADF = acid detergent fiber.

1 Values are the means of 8 replicates per treatment.

**Table 4. Effect of rumen-protected and unprotected *C. butyricum* on the ruminal fermentation of albas fattening goats.**

| Item | Group | | | SEM | P value |
|---|---|---|---|---|---|
| | CON | CB | RPCB | | |
| pH | 5.75±0.38c | 5.93±0.37b | 6.13±0.89a | 0.04 | <0.01 |
| Ammonia-nitrogen, mg/dL | 26.80±3.30 | 30.49±2.03 | 31.84±1.03 | 1.41 | 0.291 |
| Total VFA, mmol/L | 76.38±3.84 | 67.93±4.44 | 62.57±3.30 | 5.50 | 0.054 |
| Acetate, mmol/L | 47.15±1.46 | 43.47±2.26 | 41.93±3.16 | 1.86 | 0.335 |
| Propionate, mmol/L | 20.69±1.13a | 18.32±0.78a | 13.29±0.95b | 1.07 | <0.01 |
| Butyrate, mmol/L | 8.54±1.44 | 7.68±0.99 | 5.80±0.63 | 0.82 | 0.199 |

1 Values are the means of 4 replicates per treatment.

a-c Values within a row with different superscripts differ significantly at P<0.05.

72.11% of the OTUs in the CON, CB, and RPCB groups, respectively, whereas Bacteroides accounted for 18.0%, 19.7%, and 16.7%, respectively (Fig 1C).

To evaluate potential differences in bacterial diversity between treatments, we performed sequence alignment and estimated α and β diversity. The results revealed no significant differences in the Shannon, observed, Chao1, or Simpson indices ($P=0.315$, $P=0.343$, $P=0.343$, and $P=0.287$, respectively) (Fig 1A). The unweighted principal coordinate analysis (PCoA) plot demonstrated separation of the CON and CB groups (Fig 1B). These findings suggest that rumen-protected and unprotected *C. butyricum* treatments may influence the microbial community structure of the digestive system to a limited extent.

To investigate the effects of treatment on the fecal microbiota, the Mann–Whitney U test was performed at different classification levels. At the phylum level, no significant differences were observed between the groups ($P>0.05$). At the genus level, 18 genera significantly differed. Specifically, compared with those in the CON group, *Haemophilus* ($P=0.0013$), *Fusicatenibacter* ($P=0.0024$), *Butyricimonas* ($P=0.0318$), and *Actinobacillus* ($P=0.0318$) were significantly lower in both treatment groups. In addition, the CB group presented a specific reduction in *Flavonifractor* ($P=0.0318$) and *Ruminiclostridium_1* ($P=0.0424$), whereas the RPCB group presented a significant increase in *DNF 00809* ($P=0.006$), *Ruminiclostridium_6* ($P=0.0318$), and *Eubacterium nodatum_group* ($P=0.0493$) but a significant decrease in *Erysipelotrichaceae UCG-003* ($P=0.0415$) (Fig 2). These findings suggest that both the rumen-protected and unprotected *C. butyricum* treatments can alter the fecal microbiota composition, with varying effects on specific genera.

### 3.5. Antioxidant and immune function

To investigate the potential impact of dietary treatment on immune function, an intraperitoneal injection of LPS was administered to the goats, and blood samples were collected to evaluate changes in immune and antioxidant parameters among the groups. The results indicated that there was no statistically significant difference in immune or antioxidant parameters, except for IgM, among the groups ($P>0.05$). Specifically, after LPS injection, the IgM levels in the CON and RPCB groups decreased, whereas compared with those in the C ON and RPCB groups, the IgM levels in the CB group significantly increased ($P<0.01$) (Fig 3). These findings suggest that dietary supplementation with *C. butyricum* may have a positive effect on the immune response of goats in the context of LPS challenge.

We conducted a metabonomic analysis of the plasma samples via an LC–MS method, which allowed us to quantify and identify 545 metabolites. These metabolites belong to different classes, such as lipids and lipid-like molecules, benzenoids, organic acids and their derivatives, and *organoheteric* compounds. We identified these metabolites as being involved in approximately 84 different metabolic pathways, including glycine and serine metabolism (12 metabolites), alanine metabolism (5), bile acid biosynthesis (11), arginine and proline metabolism (9), ammonia recycling (6), and glutamate metabolism (8).

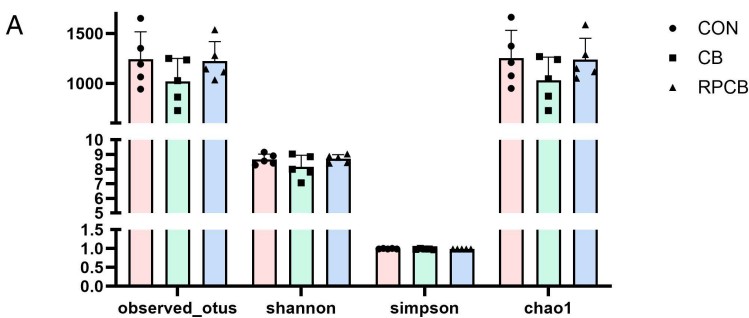

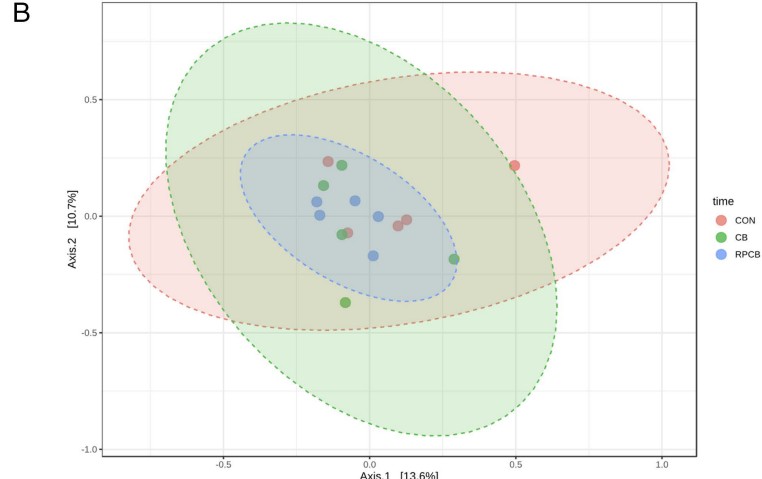

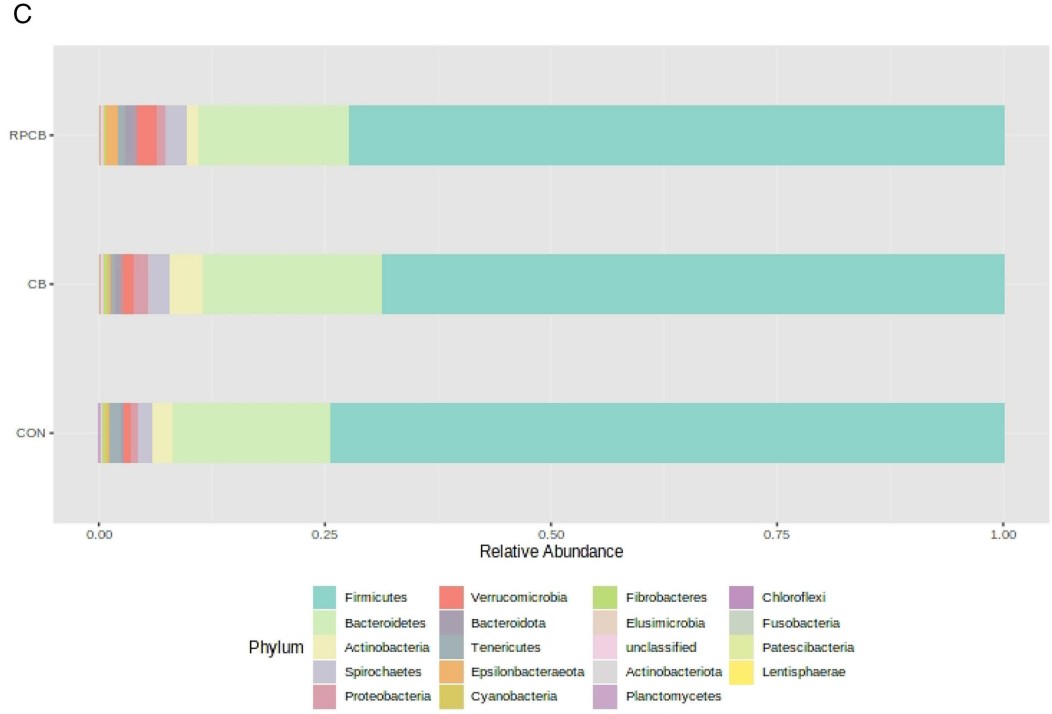

**Fig 1. Fecal microbiome diversity and structure analysis.** (A) Species diversity differences among the 3 groups were estimated by the observed species and the Shannon, Simpson, and Chao1 indices. (B) PCoA plot base of the relative abundance of OTUs (97% similarity level) showing bacterial structural clustering. (C) Proportion of bacterial phyla in each group; n = 5 for each group.

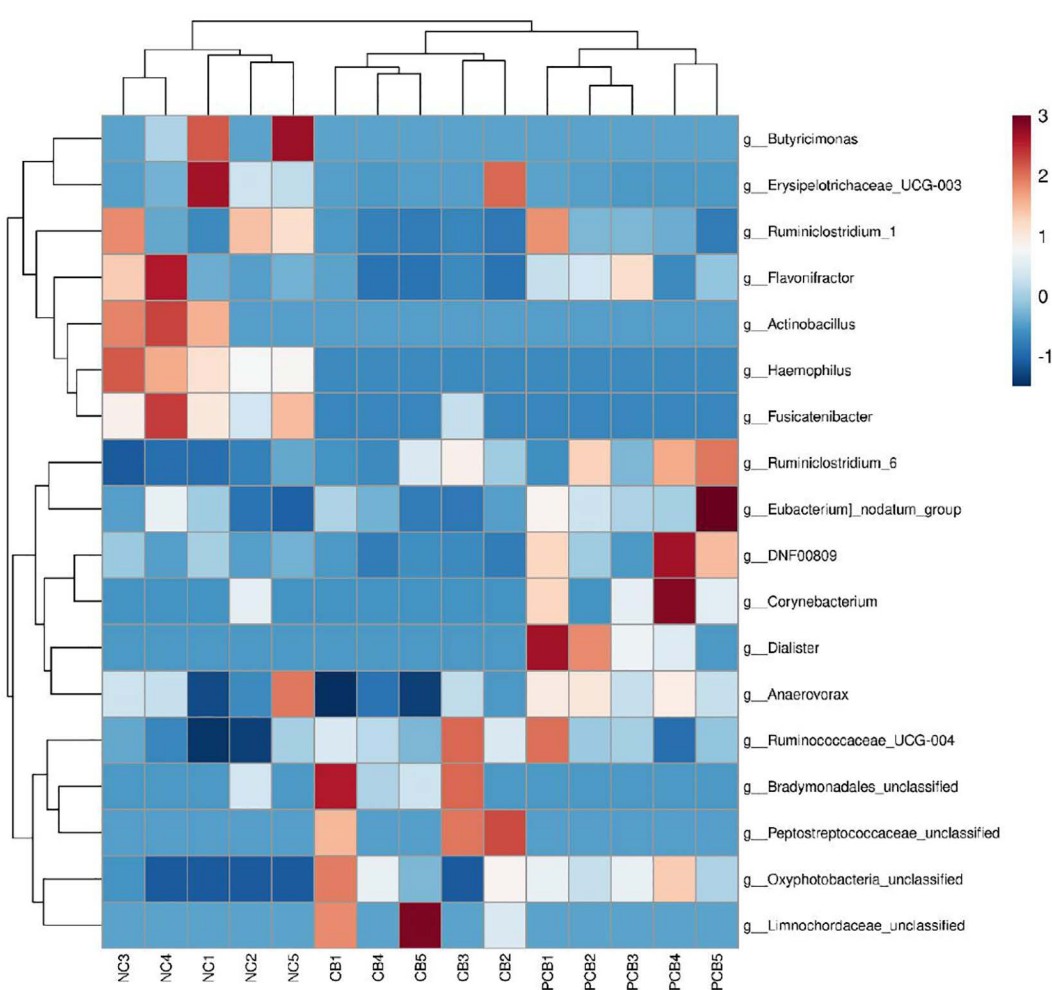

**Fig 2. Heatmap of the relative abundances of the 18 genera (97% similarity level) that differentiated the 3 groups.** The genera are shown from lower abundance (in blue) to higher abundance (in red) for the z-transformed data (n = 3 per group). The data were analyzed via the Wilcoxon rank-sum test (Mann–Whitney U test).

Compared with those in the CB group, lipids and lipid-like molecules (fatty acyls) and organoheterocystic compounds (azoles) among the metabolites in the RPCB group were increased, whereas compared with those in the CON group, lipids and lipid-like molecules, organic acids and derivatives, organic oxygen compounds, and organoheterocystic compounds were increased. Compared with those in the CON group, the contents of lipids and lipid-like molecules (fatty acyls) and organic acids and their derivatives (carboxylic acids and derivatives) in the CB group increased, whereas the contents of organic oxygen compounds (organooxygen compounds) decreased. The levels of 8-amino-7-oxononanoate,

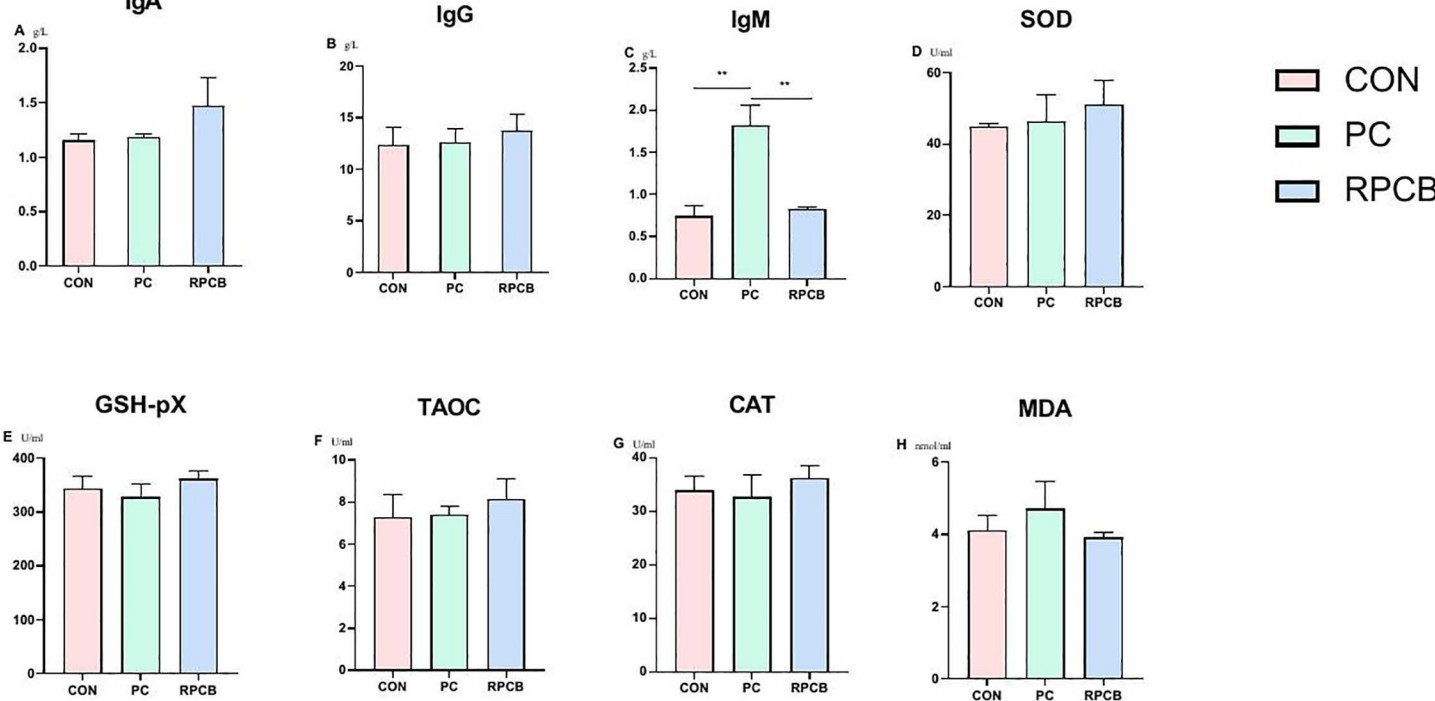

**Fig 3. Effects of rumen-protected and unprotected *C. butyricum* on plasma metabolites after LPS challenge.** Concentrations of IgA (A), IgG (B), IgM (C), SOD (D), GSH-PX (E), TAOC (F), CAT (G), and MDA (H) in plasma from each group (n = 3 per group). Mixed-effects linear regression models were used; *, P < 0.05; **, P < 0.01.

decanamide, and hexanoylglycine in the two groups of *C. butyricum* were greater than those in the CON group, but there was no significant difference between the two groups. In the CB group, the levels of 3-(1,1,2,3,3,3-hexafluoropropylene) adamantane-1-carboxylic acid and botrydial were significantly lower than those in the other two groups. The amount of tetradecanedioic acid in the RPCB group was significantly greater than that in the other two groups.

The projection (VIP) values of the ions and volcanic maps (*VIP* > 2, *P* < 0.05, *FC* > 2) were calculated via the OPLS-DA model to identify differentially abundant metabolites (Fig 4). Compared with the CB group, the RPCB group presented an increase in lipids, lipid-like molecules (*fatty acyls) and organoheterocyclic* compounds (azoles) among the metabolites. On the other hand, compared with those in the CON group, an increase in lipids and lipid-like molecules, organic acids and derivatives, organic oxygen compounds, and *organoheterocyclic* compounds was observed. Compared with the CON group, the CB group presented an increase in lipids and lipid-like molecules (fatty acyls) and organic acids and derivatives (carboxylic acids and derivatives) but a decrease in organic oxygen compounds (organooxygen compounds). The levels of 8-amino-7-oxononanoate, *decanamide*, and *hexanoylglycine* in the *C. butyricum*-treated groups were greater than those in the CON group, although no significant differences were detected between the two groups. Compared with those in the other two groups, the levels of 3-(1,1,2,3,3,3-hexafluoropropylene) adamantane-1-carboxylic acid and *botrydial* in the CB group were significantly lower. Additionally, the *tetradecanedioic* acid level was significantly greater in the RPCB group than in the other two groups.

When differentially abundant metabolites were entered into the KEGG database, 11 KEGG sequence numbers were determined. When the differentially abundant metabolites were imported into the KEGG database, 2 pathways were identified. The application of MetaboAnalyst *5.0* software enabled an in-depth analysis of the differentially abundant metabolites, revealing a significant effect of RPCB treatment on the primary bile acid biosynthesis and taurine and hypotaurine metabolism pathways compared with those in the CON group (*P* < 0.05) (Table 5).

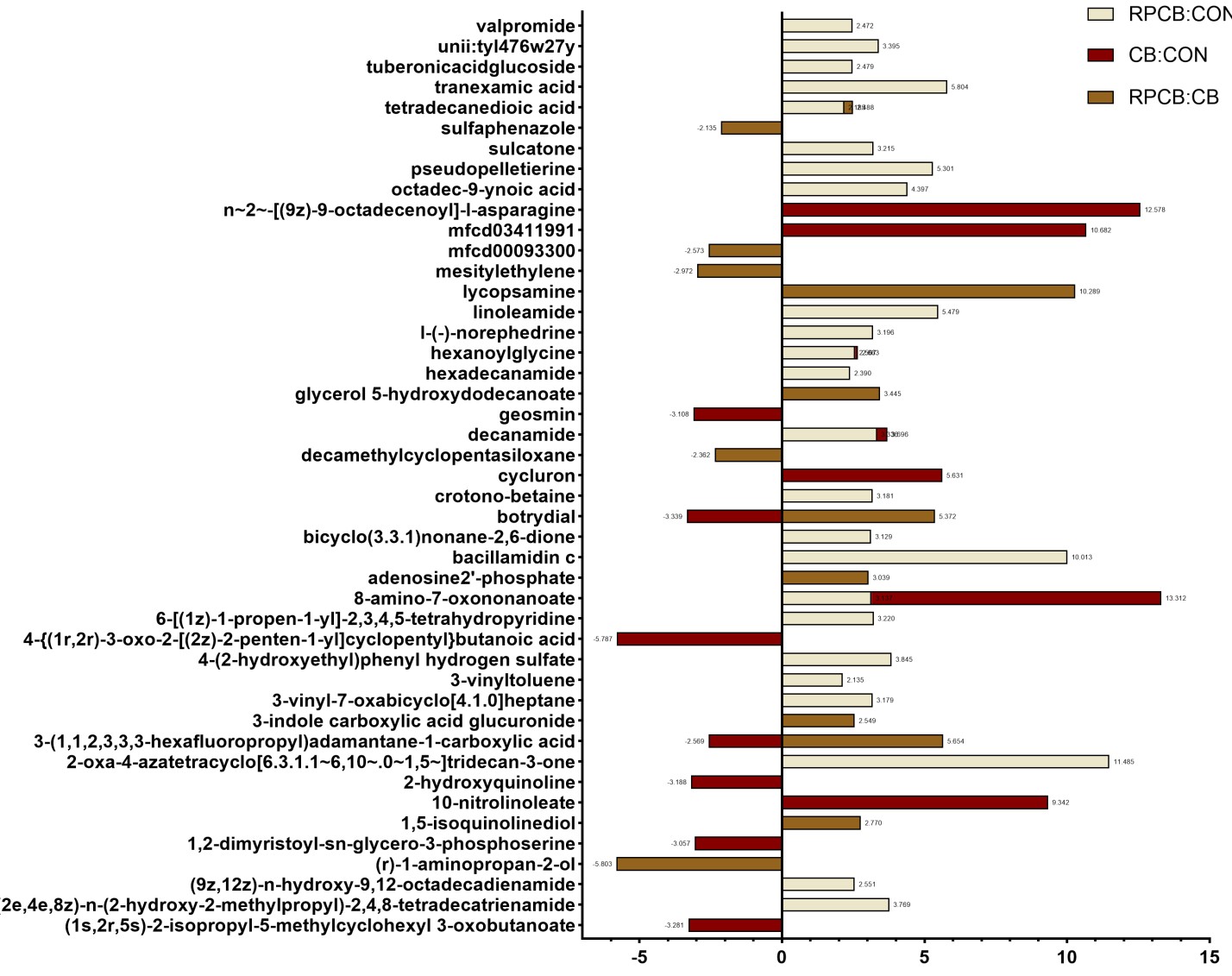

**Fig 4. Differences in metabolites in the plasma of Albas fattening goats fed rumen-protected and unprotected *C. butyricum*.** The metabolites in the plasma of Albas fattening goats fed rumen-protected and unprotected *C. butyricum* were analyzed through LC–MS analysis. The metabolites in the plasma were analyzed by LC–MS analysis, and the significance of *P*<0.05 and *VIP*>2.0 (n=3 per group) was used to estimate the filtering of different metabolites. Folding changes (FCs) indicate relative amounts.

**Table 5. Enrichment of the KEGG pathway using significantly changed metabolites from the combined LC–MS analyses.**

| Sample | KEGG Pathway name | -log(P value)[1] | Impact value[2] |
|---|---|---|---|
| RPCB: CON | Taurine and hypotaurine metabolism | 0.0314 | 0.429 |
| RPCB: CON | Primary bile acid biosynthesis | 0.170 | 0.0230 |

[1]KEGG pathway P values were calculated by comparing the proportions of metabolites in each pathway.

[2]The impact value is calculated by adding up the importance measures of each of the matched metabolites and then dividing by the sum of the important measures of all metabolites in each pathway.

## 3.6. Correlation analysis

On the basis of the fecal microbiome and metabolomic data obtained, Spearman correlation cluster analysis was performed to investigate the associations between the fecal microflora and related metabolites affected by rumen-protected and unprotected *C. butyricum* (Fig 5A). The correlations between microorganisms and metabolites were evaluated, and the results are shown in Fig 5B. Notably, a significant correlation was observed between *Haemophilus* and other

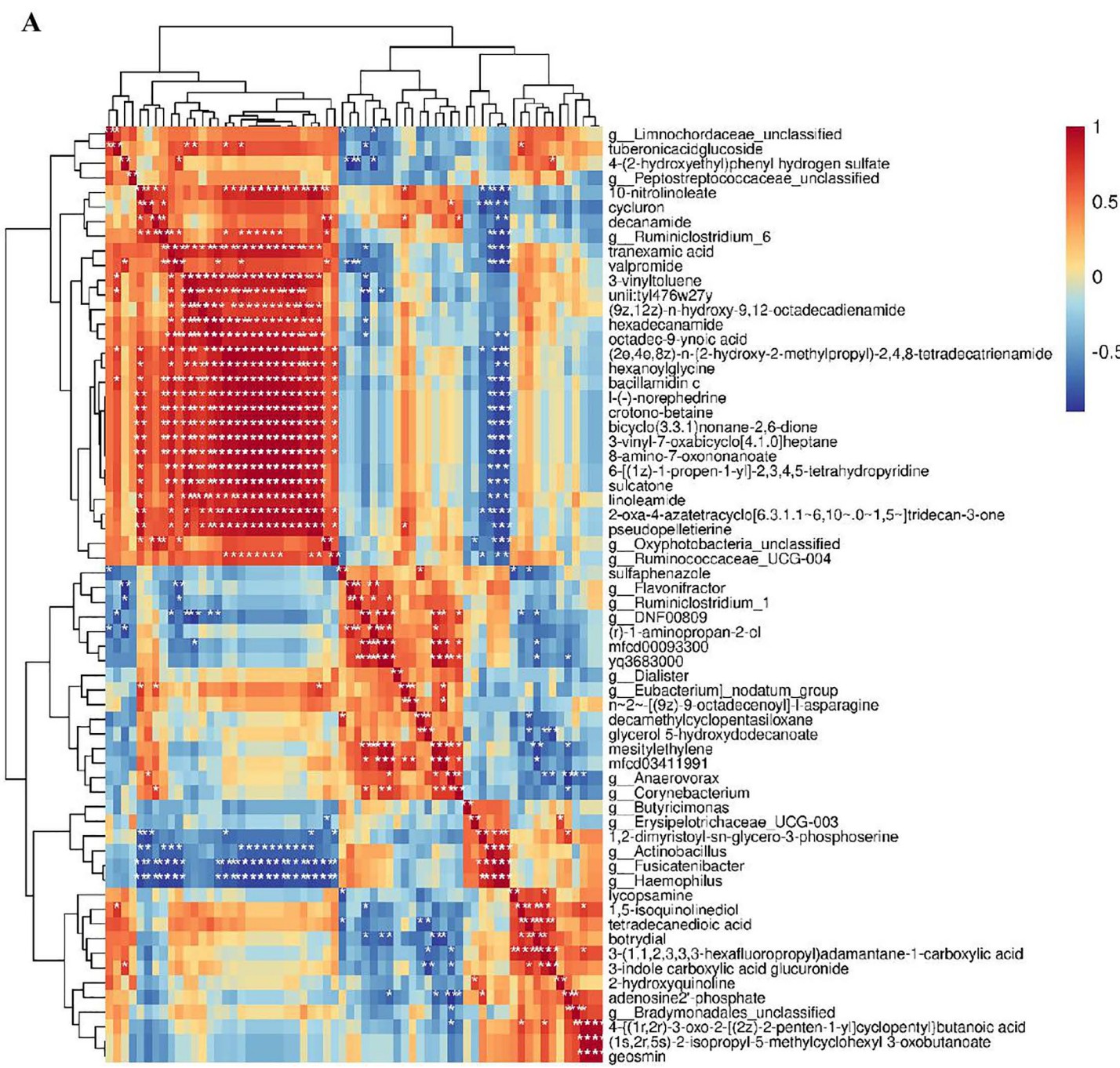

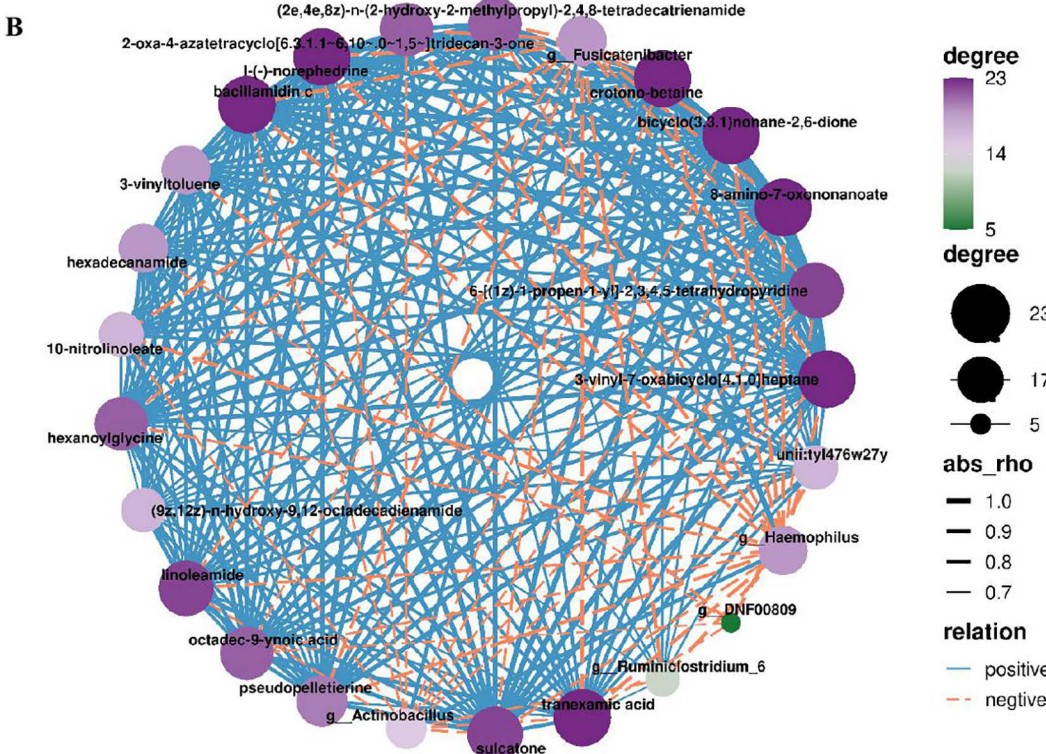

**Fig 5. Integrated correlation-based network analysis of microbes and metabolites. Pearson's correlation analysis of the entire network in the 3 groups.** (A) Comprehensive correlation heatmap analysis of microorganisms and metabolites. (B) Integrated correlation-based network analysis (Pearson's correlation) of microbes and metabolites.

differential bacteria (at the genus level) and differentially abundant metabolites in feces. In addition, compared with those in the CON group, *Haemophilus* ($P = 0.0013$) and *Actinobacillus* ($P = 0.0318$) were significantly lower in both treatment groups. A decrease in the abundance of *Haemophilus* is beneficial for improving the immune capacity of the host intestine. These findings suggest a potential link between fecal microorganisms and blood metabolism and suggest that the administration of *C. butyricum* may alleviate intestinal inflammation by modulating the species of the intestinal flora and thereby influencing the composition of metabolic products.

## 4. Discussion

To the best of our knowledge, no previous studies have investigated whether rumen-protected *C. butyricum* has superior growth-promoting and blood immune-enhancing effects than unprotected *C. butyricum* does. Our findings indicate that there was no significant difference in growth performance or fecal microflora structure among the three groups. However, both *C. butyricum* treatments reduced the content of harmful bacteria and had the potential to improve fatty acid metabolism and immune performance. Unprotected *C. butyricum* is more conducive to immune activation, whereas protected *C. butyricum* can effectively reduce the propionic acid content and regulate the pH of the rumen.

Probiotics play a significant role in regulating host health by improving nutrient utilization and immune responses and altering the composition of the gut microbiota [33]. The long-term administration of probiotics has been shown to promote resistance to bacterial invasion, reduce inflammation, and alleviate stress [11]. Studies have confirmed that *C. butyricum* can regulate the intestinal flora of monogastric animals to enhance growth performance, immunity, and antioxidation

[34,35]. To investigate the potential impact of *C. butyricum* on the production performance of ruminants, goats were fed a diet supplemented with either $1 \times 10^9$ CFU/kg unprotected *C. butyricum* or rumen-protected *C. butyricum*. However, neither supplementation significantly affected the growth performance nor apparent digestibility of the goats. The addition of 0.05% *C. butyricum* to the diet has been reported to improve rumen fermentation efficiency and promote growth in some studies [20], whereas others have reported no significant effect [26]. This variation may be attributed to environmental differences or the use of different strains of *C. butyricum*. Evidence suggests that probiotics may have limited effects on the growth performance of postweaning ruminants and that the effects of the same strain on ruminants may depend on the dose and age of the animals [36–38]. The C. butyricum concentration has also been reported to influence the effect of *C. butyricum* on ruminants in previous studies [26]. Therefore, the concentration of *C. butyricum* may also be a determining factor in the production performance of ruminants.

Rumen-protected and unprotected *C. butyricum* have been investigated for their potential to alleviate the decrease in rumen pH under high-concentrate diet conditions. These treatments reduce the concentration of propionic acid in the rumen, thereby preventing liver burden and meat quality degradation [39]. Notably, the rumen pH in the RPCB group was significantly greater than that in the CB group. These findings underscore the potential of rumen-protected and unprotected *C. butyricum* in maintaining optimal rumen pH and improving meat quality under high-concentrate diet conditions. *C. butyricum* is a starch-fermenting bacterium that competes with other bacteria that ferment starch in the rumen, leading to a decrease in the concentration of propionic acid, as previously reported [26,40]. Another reason for this decrease may be the production of butyric acid by unprotected *C. butyricum*, which can inhibit the metabolism of propionic acid in the rumen [41]. Additionally, the current study revealed that rumen-protected fat can facilitate the entry of *C. butyricum* into the intestine to digest escaped starch in the rumen. Moreover, butyric acid, a metabolite of *C. butyricum*, was found to enhance the growth performance of weaned ruminants by increasing the length of the intestinal villi and improving the rumen epithelium [15,42,43]. It is hypothesized that both rumen-protected and unprotected *C. butyricum* may modulate the intestinal microflora, regulate intestinal epithelial cells, and secrete active compounds into the systemic circulation, which in turn can stimulate rumen function, influence the utilization of volatile fatty acids (VFAs) in the rumen, particularly propionic acid, and thereby regulate the pH of the rumen.

We conducted a study using LC–MS metabonomic methods to analyze plasma samples and reported that the administration of rumen-protected and unprotected *C. butyricum* led to significant changes in blood metabolite levels. We detected complex changes in fatty acyls and observed that the number of Azoles was greater in the RPCB group than in the CB group. Additionally, carboxylic acids and derivatives, as well as *organooxygenic* compounds, were increased in both the CB and RPCB groups compared with those in the CON group. Furthermore, 3-(1,1,2,3,3,3-hexafluoropropylene) adamantane-1-carboxylic acid (carboxylate) was abundant only in the blood of the CB group, whereas *tetradecanedioic* acid (bile acid intermediate) was overexpressed only in the blood of the RPCB group. Compared with those in the CON group, the concentrations of *hexanoylglycine* in the blood of both the CB and RPCB groups were elevated, suggesting an increase in β-oxidation, which could be due to the increased initial substrate carboxylate content in the CB group and the increased intermediate products in the RPCB group. Both treatments elevated the levels of acetyl coenzyme A produced by β-oxidation and subsequently synthesized *hexanoylglycine* with glycine, which was then transported to extrahepatic tissues such as muscles and kidneys for utilization [44,45]. This process promotes fat decomposition and results in increased energy entering the body's circulation. The insignificant improvement in growth performance may be attributed to this increased fat decomposition [46]. These findings suggest that feeding ruminants with protected or unprotected *C. butyricum* increases β-oxidation and leucine catabolism in the liver, which may affect weight gain in goats.

Alterations in the gastrointestinal tract microbiota can significantly impact the digestive and immune functions of the host. The administration of unprotected *C. butyricum* in the intestine may modulate the gut microbial ecology, reinforce the intestinal barrier and confer immunomodulatory benefits. Similarly, rumen-protected C. butyricum exerts analogous effects on the hindgut barrier as unprotected *C. butyricum* does; however, it also enriches some uncategorized bacterial taxa

[27]. In concurrence with these findings, this study demonstrated that both rumen-protected and unprotected *C. butyricum* reduce the abundance of detrimental bacteria (*Haemophilus* [47], *Butyriconas* [48], Actinobacillus) in the feces. Both formulations of *C. butyricum* may attenuate the pathogen burden in the gastrointestinal tract, thereby increasing the immune competence of the host.

The aim of this study was to investigate the impact of rumen-protected and unprotected *C. butyricum* on the immunity and antioxidation of ruminants, with a particular focus on the effects on blood, using LPS challenge as a means to simulate acute inflammation. LPS is a toxin generated by *Escherichia coli* that triggers an immune response and elevates the levels of inflammatory factors and peroxides when it enters the body through a damaged digestive tract [34]. To evaluate the immune and antioxidant response, serum samples were collected 6 hours after the intraperitoneal injection of 1 μg/kg LPS. Our results showed that, unlike monogastric animals, long-term feeding of rumen-protected or unprotected *C. butyricum* did not significantly affect the immune and antioxidant indices of healthy goats. Six hours after the intraperitoneal injection of LPS and subsequent collection of serum samples, a significant increase in the IgM levels of the CB group was observed compared with those of the other two groups. IgM is the first antibody produced by the immune system in response to infection or immunization and is known to reduce the inflammatory effects of LPS treatment and alleviate LPS toxicity [49]. The higher levels of IgM in the CB group may have promoted the activation of the immune system, thereby playing a preventive role.

In this study, no positive effects of metabolites of *C. butyricum* on immune or antioxidant metabolic pathways were detected, but rumen-protected *C. butyricum* significantly promoted taurine and *hypotaurine* metabolism in the liver. Taurine can compete with GSH for substrates in metabolic pathways, act on cells to reduce LPS-induced ROS generation and apoptosis, and has antioxidant effects [50,51]. Taurine can prevent GABA transaminase from increasing the GABA concentration and can also bind to GABAA receptors to mimic the effects of GABA. Midazolam also affects postsynaptic G-aminobutyric acid type A (GABAA) receptors, thereby exerting an anti-inflammatory effect on colitis. It also interferes with GABA reuptake, enabling endogenous acetylcholine to increase GABA release, improve cell viability, and exert anti-inflammatory effects to reduce apoptosis [52,53]. However, blood midazolam concentrations decreased in the RPCB group, counteracting the increase in taurine. Therefore, the reduction in Midazolam may inhibit the anti-inflammatory and antioxidant capacity of *C. butyricum* in the host. Moreover, the concentration of tranexamic acid in the RPCB group also increased, improving the coagulation mechanism and exerting anti-inflammatory effects [44].

Previous studies have shown that the enzyme that converts KAPA to biotin is missing in *C. butyricum* [54]. KAPA may be used as a biomarker for the body's utilization of *C. butyricum* metabolites. The high levels of KAPA in the blood of both the CB and RPCB groups in the present study may support this point of view. The higher KAPA concentration in the RPCB group may indicate that more *C. butyricum* metabolites are absorbed in the intestine and that the rumen-protected *C. butyricum* utilizes the same volatile acid in the rumen as the unprotected *C. butyricum.*

In summary, both interventions were effective in reducing the prevalence of pathogenic bacteria in feces, increasing β-oxidation products, potentially improving immunity and enhancing fatty acid metabolism in the liver, but the antioxidant capacity did not change significantly. Upon stimulation with LPS, feeding with unprotected C. butyricum significantly increased IgM expression, suggesting a positive impact on the immune response. This research provides a basis for further exploration of the mechanisms underlying the effects of *C. butyricum* on ruminants.

## Author contributions

**Conceptualization:** Caixia Zhang, Jiyu Han, Hairong Wang.

**Formal analysis:** Caixia Zhang.

**Funding acquisition:** Jiyu Han, Hairong Wang.

**Investigation:** Jiyu Han.

**Methodology:** Caixia Zhang.

**Software:** Jiyu Han.

**Supervision:** Hairong Wang.

**Writing – original draft:** Caixia Zhang.

**Writing – review & editing:** Hairong Wang.

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
