## [Editor Report · Decision Letter 0]

12 Jan 2025

Dear Dr. Wang,

Thank you for submitting your manuscript to PLOS ONE. After careful consideration, we feel that it has merit but does not fully meet PLOS ONE’s publication criteria as it currently stands. Therefore, we invite you to submit a revised version of the manuscript that addresses the points raised during the review process.

Dear Authors, While the manuscript demonstrates significant effort and scientific value, it requires substantial improvement in areas such as experimental justification, clarity of results, and interpretation. 

Few points should be addressed before the review process 

We look forward to receiving your revised manuscript.

Kind regards,

Aziz ur Rahman Muhammad

Academic Editor

PLOS ONE

Journal Requirements:

2. In your Methods section, please provide additional details regarding participant consent from the owners of the animals. In the ethics statement in the Methods and online submission information, please ensure that you have specified (1) whether consent was informed and (2) what type you obtained (for instance, written or verbal). If the need for consent was waived by the ethics committee, please include this information.

3. Thank you for stating the following financial disclosure: This research was funded by the National Key Research and Development Program of China (2022YFD1602305-03), Inner Mongolia Department of Science and Technology (2023YFDZ0004), National Center of Technology Innovation for Dairy, Research Breakthroughs (2022-1).

Additional Editor Comments:

Dear Authors, While the manuscript demonstrates significant effort and scientific value, it requires substantial improvement in areas such as experimental justification, clarity of results, and interpretation.

Few points should be addressed before the review process

In abstract section, avoid redundancy, such as repeating "rumen-protected and unprotected C. butyricum" and statistical results (P-values) should be included for all key findings mentioned in the abstract. In introduction section, please provide potential mechanisms through which C. butyricum affects immune performance and include more recent references from 2020–2024 to contextualize the study. Furthermore, hypotheses should be clearly stated in introduction section and objectives should be based on strong hypotheses. In result section, highlight specific changes in microbial genera that are relevant to immunity or antioxidation. While in discussion section, do not overstates conclusions in some areas. For example, avoid claiming "better immune protection" without stronger statistical backing. Please also explain potential mechanisms for the observed effects (or lack thereof) of the treatments. In the last the legends of table and figure should provide more detailed explanations, especially regarding abbreviations and statistical notations. Please also include the number of replicates (N) in all figure legends.

---

## [Author Response · Author response to Decision Letter 1]

27 Mar 2025

Dear Reviewer:

We are truly grateful for your critical comments and thoughtful suggestions for our manuscript. These comments and suggestions were helpful, and we have revised the manuscript carefully. The revised portions have been marked in red in the revised manuscript. Our point-by-point responses to your comments/suggestions are provided below.

Comment 1: In abstract section, avoid redundancy, such as repeating "rumen-protected andunprotected C. butyricum" and statistical results (P values) should be included forall key findings mentioned in the abstract.

Response 1:

Thank you for your suggestion. I apologize for the unnecessary misunderstanding caused by our mistakes. On the basis of the suggestions and guidance from you and the other reviewers, we have modified the repetitive expressions and supplemented the statistical results in the revised manuscript. (Lines 11--32)

Comment 2: In introduction section, please providePotential mechanisms through which C. butyricum affects immune performanceand include more recent references from 2020-2024 to contextualize the study.

Response 2:

Thank you for your suggestion. The revised manuscript has been supplemented with information on the potential mechanism of action of C. butyricum in enhancing immune function and has updated the references. (Lines 46--61)

Comment 3: Furthermore, hypotheses should be clearly stated in introduction section and objectives should be based on strong hypotheses.

Response 3:

Thank you for your suggestion. We have rewritten the section in question. (Lines 62--69)

Comment 4: In result section, highlightspecific changes in microbial genera that are relevant to immunity or antioxidation.

Response 4:

Thank you for your suggestion. In the revised version, the specific changes in the microorganisms of mice related to immune function have been supplemented according to your suggestions. (Lines 319--331)

Comment 5: For example, avoid claiming "better immune protection" without stronger statistical backing. Please also explain potential mechanisms for the observed effects (or lack thereof) of the treatments.

Response 5:

Thank you for your suggestion. According to your suggestions, the expressions of some conclusions have been changed in the revised manuscript. (Lines 448--456)

Six hours after the intraperitoneal injection of LPS and the collection of serum samples, the IgM level in the CB group was significantly greater than that in the other two groups. Currently, we do not have a very clear mechanism to explain this phenomenon. In subsequent research, we will focus on potential mechanisms in this regard.

Comment 6: In the last the legends of table and figure shouldprovide more detailed explanations, especially regarding abbreviations andstatistical notations. Please also include the number of replicates (N) in all figure legends.

Response 6:

Thank you for your suggestion. According to your suggestion, provide a more detailed explanation of the legends for both the tables and the figures, and indicate the number of repetitions (N) in all the legends. (Lines 245--248, 298--303)

---

## [Decision Letter · Decision Letter 1]

26 May 2025

Dear Dr. Wang,

Thank you for submitting your manuscript to PLOS ONE. After careful consideration, we feel that it has merit but does not fully meet PLOS ONE’s publication criteria as it currently stands. Therefore, we invite you to submit a revised version of the manuscript that addresses the points raised during the review process.

**Dear Authors**

**Thank you for revising the manuscript as suggested by me. Now reviewer suggested revisions, therefore, i am inviting you to revise the manuscript as suggested by reviewers. Please address reviewer’s comments especially statistical part of the manuscript.**

We look forward to receiving your revised manuscript.

Kind regards,

Aziz ur Rahman Muhammad

Academic Editor

PLOS ONE

**Additional Editor Comments:**

Dear Authors

Thank you for revising the manuscript as suggested by me. Now reviewer suggested revisions, Therefore, i am inviting you to revise the manuscript as suggested by reviewers. Please address reviewers comments specially statistical part of the manuscript.

Reviewers' comments:

Reviewer's Responses to Questions

**Comments to the Author**

Reviewer #1: All comments have been addressed

Reviewer #2: All comments have been addressed

Reviewer #3: (No Response)

2. Is the manuscript technically sound, and do the data support the conclusions?

Reviewer #1: Yes

Reviewer #2: Yes

Reviewer #3: No

3. Has the statistical analysis been performed appropriately and rigorously?

Reviewer #1: Yes

Reviewer #2: Yes

Reviewer #3: No

4. Have the authors made all data underlying the findings in their manuscript fully available?

Reviewer #1: Yes

Reviewer #2: Yes

Reviewer #3: Yes

5. Is the manuscript presented in an intelligible fashion and written in standard English?

Reviewer #1: Yes

Reviewer #2: Yes

Reviewer #3: No

**Reviewer #1: ** My comments were thoroughly addressed by the authors, and I appreciate the careful attention given to the points I raised in the previous round of review. The revised manuscript demonstrates substantial improvement in clarity, scientific rigor, and overall presentation. All of the issues and suggestions previously mentioned have been satisfactorily resolved, and I find no remaining concerns that would require further revision at this stage. Therefore, I am satisfied with the current version of the manuscript and have no additional comments or suggestions. I recommend that the manuscript be accepted for publication in its present form.

**Reviewer #2:**  Thank you for the opportunity to review the manuscript titled "Effects of rumen-protected or unprotected Clostridium butyricum on growth performance, rumen fermentation, immunity and antioxidation in fattening goats." The study gives important insights into using rumen-protected or unprotected Clostridium butyricum on the ruminal fermentation and growth performance of goats. The study is well designed, the methodology is appropriate, and the results are clear and well laid out. I just have a minor comment: The introduction needs to be improved, and English language editing might be required.

**Reviewer #3: ** I have reviewed the manuscript “Effects of rumen-protected or unprotected Clostridium butyricum on growth performance, rumen fermentation, immunity and antioxidation in fattening goats”, and found it interesting. I have provided several comments on current study, especially to improve the quality and statistical analysis. If authors agrees to revise the manuscript as I suggested, please accept the manuscript, otherwise reject it.

Abstract

Line 15: please correct rumen protected and rumen unprotected in sentence ‘…effects of rumen protection with or without C. butyricum’

Line 23-24: these are results ‘thus alleviating the decrease in rumen 24 pH’ not discussion please indicate it in the form of result

Line 25: please remove it ‘albeit through different mechanisms’

Line 25-27: please be specific ‘The difference in the fecal flora between the CB and RPCB groups was limited, but the content of harmful bacteria in the feces of the other two groups decreased compared with that in the CON group’ what do you mean by harmful bacteria

Line 28-30: please rewrite ‘8-Amino-7-oxononanoate (KAPA) can serve as a biomarker for the effect of C. butyricum on the body’ its not clear and write in the form of result

Some statistical outcomes are not supported by clear metrics (e.g., “greater ability to reduce propionic acid” needs clearer quantification).

Introduction:

Line 38-40: don’t write anything without reference ‘When used as feed additives, antibiotics have been shown to increase ruminant feed utilization and inhibit the growth of harmful bacteria by increasing immunity and preventing disease’ please include citation. Anas S. Dablool, Banan Atwah, Saad Alghamdi, Maha Abdullah Momenah, Ohud Saleh, Nada Alhazmi, Yasser S. Mostafa, Saad A. Alamri, Worood A.A. Alyoubi, Naheda M. Alshammari, Alaa S. Mohamed, Nadeen G. Mostafa and Belal A. Omar. Could Paenibacillus xylanexedens MS58 be an Ecofriendly Antibiotic in Poultry Production? Impacts on Performance, Blood Biochemistry, Gut Microbiota and Meat Quality.Pak Vet J, 2024, 44(2): 352-360. And Fatimah S. Alqahtani, Safia M.A. Bahshwan, Mada M. AL-Qurashi, Aminah Allohibi, Eman A. Beyari, Mashail A. Alghamdi, Roua S. Baty, Nawal Al-Hoshani, Anas S. Dablool, Fadwa Mohammed Alkhulaifi, Abdullateef A. Alshehri, Essam H. Ibrahim,0, Ahmed M. Saad, Nadeen G. Mostafa

Impact of Dietary Bacillus toyonensis M44 as an Antibiotic Alternative on Growth, Blood Biochemical Properties, Immunity, Gut Microbiota, and Meat Quality of IR Broilers. Pak Vet J, 2024, 44(3): 637-646

Line 40-43: ‘However, the widespread use of antibiotics in animal feed has raised concerns over the accumulation of drug residues in the environment and animal products, which in turn contributes to the development of antibiotic-resistant bacteria and poses a risk to human health [2]’ citation is outdated nearly 10 year back. Please include recent citation. The citation could be ‘Rida Haroon Durrani, Ali Ahmad Sheikh, Muhammad Humza, Salman Ashraf, Aleena Kokab, Tauqeer Mahmood and Muhammad Umar Zafar Khan.Evaluation of Antibiotic Resistance Profile and Multiple Antibiotic Resistance Index in Avian Adapted Salmonella enterica serovar Gallinarum Isolates. Pak Vet J, 2024, 44(4): 1349-1352’ and ‘Magnoli AP, Parada J, Luna MarÃ¬a J, Corti M, Escobar FM, FernÃ¡ndez C, Coniglio MV, Ortiz ME, Wittouck P, Watson S, Cristofolini LA and Cavaglieri L, 2024. Impact of probiotic Saccharomyces cerevisiae var. boulardii RC009 alone and in combination with a phytase in broiler chickens fed with antibiotic-free diets. Agrobiological Records 16: 1-10.’ And ‘Ezzat M, Hassanin AAI, Mahmoud AE, Ismail SM and El-Tarabili RM, 2023. Risk factors, antibiotic profile, and molecular detection of virulence and antibiotic resistance genes of enteric bacteria in diarrheic calves in Egypt. International Journal of Veterinary Science 12(2): 161-168.’ And ‘Bui MTL, Nguyen TT, Nguyen HC, Ly KLT and Nguyen TK, 2024. Antibiotic resistance and pathogenicity of Escherichia coli isolated from cattle raised in households in the Mekong Delta, Vietnam. International Journal of Veterinary Science 13(5): 730-736.’

Line 43-44: rewrite the sentence ‘Given the significant impact of livestock food production on both human health and the environment, the identification of new feed additives to replace antibiotics and enhance feed efficiency has become crucial’ and new sentence could be ‘Given the significant impact of livestock food production on both human health and the environment, the new feed additives to replace antibiotics and enhance feed efficiency are being used in livestock industry’ and also include citations ‘Abdul Hannan, Xiaoxia Du, Bakhtawar Maqbool3 and Ahrar Khan. Nanoparticles as Potent Allies in Combating Antibiotic Resistance: A Promising Frontier in Antimicrobial Therapy. Pak Vet J, 2024, 44(4): 957-967’ and ‘Maha J. Balgoon and Amira M. Alghamdi. Biochemical Assessment of Boswellic Acid Enrich-Frankincense Extract and its Antioxidant, Antibacterial, Anticancer and Anti-inflammatory Potential in Ameliorating the Glycerol-Toxicity in Rats. Pak Vet J, 2024, 44(4): 1023-1032’ and ‘Mohamed A. Elzaiat, Abdelrahman S. Mandour, Mohamed A. H. Youssef, Hany A. Wafa, Salma M. Aljahdali, Amani Osman Shakak, Latifa Al Husnain, Mohammed A. Alqahtani, Mashail A. Alghamdi, Amani Omar Abuzaid, Tahani M. Alqahtani, Hawazen K. Al Gheffari, Nahla Alsayd Bouqellah, Rania M.Y. Heakel Biochemical and Molecular Characterization of Five Basil Cultivars Extract for Enhancing the Antioxidant, Antiviral, Anticancer, Antibacterial, and Antifungal Activities. Pak Vet J, 2024, 44(4): 1105-1119’ and ‘Rashid S, Alsayeqh AF, Akhtar T, Abbas RZ and Ashraf R, 2023. Probiotics: Alternative of antibiotics in poultry production. International Journal of Veterinary Science 12(1): 45-53. ’

Please also provides background on C. butyricum use in livestock, but the literature review is not comprehensive and citations are outdated.

The hypothesis is vague and not well-integrated with objectives.

Unclear rationale for selecting 3 animals per group for blood collection. Also, using different subsets of animals for different measurements (e.g., n=3 for blood, n=4 for rumen pH, n=5 for microbiome) introduces variability and weakens statistical reliability.

Don’t you think use of multiple post hoc tests (LSD and Duncan) increase Type I error? How could you justify it?

Results and Discussion

General comments

Overall, this section is good, however, the authors could explain why they attempt to draw performance implications while the data is non significant. This is misleading, please correct it. In microbial composition, diversity differences are minor and reported genus-level changes (e.g., Haemophilus, Flavonifractor) lack functional interpretation and are not linked convincingly to immunity. While in immune and antioxidant markers results only IgM shows significance post-LPS injection. Other indices (IgA, IgG, T-AOC, SOD, CAT, MDA) show no changes, yet the manuscript still draws generalized conclusions, please correct it in the revised manuscirpt. I would also comment that metabolomics results contain rich data, but poorly integrated into discussion. Identified changes in plasma metabolites are not contextualized in terms of relevance to goat health or productivity. Please correct grammar and support your study with strong data while in discussion for example several statements (e.g., improvement in immune response, metabolic shifts) are unsupported by robust data and the metabolic pathway discussion lacks depth and does not explain why those pathways are relevant for goats.

**Do you want your identity to be public for this peer review?** For information about this choice, including consent withdrawal, please see our Privacy Policy

Reviewer #1: **Yes: ** Anusorn Cherdthong

Reviewer #2: No

Reviewer #3: **Yes: ** Asfa Fatima

---

## [Author Response · Author response to Decision Letter 2]

25 Jul 2025

Dear Reviewer:

We are truly grateful for your critical comments and thoughtful suggestions for our manuscript. These comments and suggestions were helpful, and we have revised the manuscript carefully. The revised portions have been marked in red in the revised manuscript. Our point-by-point responses to your comments/suggestions are provided below.

Reviewer #2:

Comment 1: The introduction needs to be improved, and English language editing might be required.

Response 1: Thank you for your suggestion. We have conducted a detailed and accurate English edit of the introduction section in accordance with your suggestions.

Reviewer #3:

Comment 1: Line 15: please correct rumen protected and rumen unprotected in sentence ‘…effects of rumen protection with or without C. butyricum’.

Response 1: Thank you for your suggestion. We have carefully revised the relevant sentences in accordance with your suggestions. (Line 15)

Comment 2: Line 23-24: these are results ‘thus alleviating the decrease in rumen pH’ not discussion please indicate it in the form of result.

Response 2: Thank you for your suggestion. We have carefully revised the relevant sentences in accordance with your suggestions. (Line 23-24)

Comment 3: Line 25: please remove it ‘albeit through different mechanisms’.

Response 3: Thank you for your suggestion. We have carefully revised the relevant sentences in accordance with your suggestions. (Line 25)

Comment 4: Line 25-27: please be specific ‘The difference in the fecal flora between the CB and RPCB groups was limited, but the content of harmful bacteria in the feces of the other two groups decreased compared with that in the CON group’ what do you mean by harmful bacteria.

Response 4: Thank you for your suggestion. ‘Harmful bacteria’ refer to those bacteria present in the intestinal tract of goats that have adverse effects on the health or physiological functions of the host. For instance, certain pathogenic strains such as Salmonella and Escherichia coli that may exist in the intestines can cause problems like intestinal infections and stunted growth in ruminants, thereby exerting negative impacts on livestock production. (Line 25-27)

Comment 5: Line 38-40: don’t write anything without reference ‘When used as feed additives, antibiotics have been shown to increase ruminant feed utilization and inhibit the growth of harmful bacteria by increasing immunity and preventing disease’ please include citation. Anas S. Dablool, Banan Atwah, Saad Alghamdi, Maha Abdullah Momenah, Ohud Saleh, Nada Alhazmi, Yasser S. Mostafa, Saad A. Alamri, Worood A.A. Alyoubi, Naheda M. Alshammari, Alaa S. Mohamed, Nadeen G. Mostafa and Belal A. Omar. Could Paenibacillus xylanexedens MS58 be an Ecofriendly Antibiotic in Poultry Production? Impacts on Performance, Blood Biochemistry, Gut Microbiota and Meat Quality.Pak Vet J, 2024, 44(2): 352-360. And Fatimah S. Alqahtani, Safia M.A. Bahshwan, Mada M. AL-Qurashi, Aminah Allohibi, Eman A. Beyari, Mashail A. Alghamdi, Roua S. Baty, Nawal Al-Hoshani, Anas S. Dablool, Fadwa Mohammed Alkhulaifi, Abdullateef A. Alshehri, Essam H. Ibrahim,0, Ahmed M. Saad, Nadeen G. Mostafa

Impact of Dietary Bacillus toyonensis M44 as an Antibiotic Alternative on Growth, Blood Biochemical Properties, Immunity, Gut Microbiota, and Meat Quality of IR Broilers. Pak Vet J, 2024, 44(3): 637-646.

Response 5: Thank you for your suggestion. We have added relevant content in accordance with your suggestions and marked the citations. (Line 38-40)

Comment 6: Line 40-43: ‘However, the widespread use of antibiotics in animal feed has raised concerns over the accumulation of drug residues in the environment and animal products, which in turn contributes to the development of antibiotic-resistant bacteria and poses a risk to human health [2]’ citation is outdated nearly 10 year back. Please include recent citation. The citation could be ‘Rida Haroon Durrani, Ali Ahmad Sheikh, Muhammad Humza, Salman Ashraf, Aleena Kokab, Tauqeer Mahmood and Muhammad Umar Zafar Khan. Evaluation of Antibiotic Resistance Profile and Multiple Antibiotic Resistance Index in Avian Adapted Salmonella enterica serovar Gallinarum Isolates. Pak Vet J, 2024, 44(4): 1349-1352’ and ‘Magnoli AP, Parada J, Luna MarÃ¬a J, Corti M, Escobar FM, FernÃ¡ndez C, Coniglio MV, Ortiz ME, Wittouck P, Watson S, Cristofolini LA and Cavaglieri L, 2024. Impact of probiotic Saccharomyces cerevisiae var. boulardii RC009 alone and in combination with a phytase in broiler chickens fed with antibiotic-free diets. Agrobiological Records 16: 1-10.’ And ‘Ezzat M, Hassanin AAI, Mahmoud AE, Ismail SM and El-Tarabili RM, 2023. Risk factors, antibiotic profile, and molecular detection of virulence and antibiotic resistance genes of enteric bacteria in diarrheic calves in Egypt. International Journal of Veterinary Science 12(2): 161-168.’ And ‘Bui MTL, Nguyen TT, Nguyen HC, Ly KLT and Nguyen TK, 2024. Antibiotic resistance and pathogenicity of Escherichia coli isolated from cattle raised in households in the Mekong Delta, Vietnam. International Journal of Veterinary Science 13(5): 730-736.’

Response 6: Thank you for your suggestion. We have attached the latest citations in accordance with your suggestions. (Line 40-43)

Comment 7: Line 43-44: rewrite the sentence ‘Given the significant impact of livestock food production on both human health and the environment, the identification of new feed additives to replace antibiotics and enhance feed efficiency has become crucial’ and new sentence could be ‘Given the significant impact of livestock food production on both human health and the environment, the new feed additives to replace antibiotics and enhance feed efficiency are being used in livestock industry’ and also include citations ‘Abdul Hannan, Xiaoxia Du, Bakhtawar Maqbool3 and Ahrar Khan. Nanoparticles as Potent Allies in Combating Antibiotic Resistance: A Promising Frontier in Antimicrobial Therapy. Pak Vet J, 2024, 44(4): 957-967’ and ‘Maha J. Balgoon and Amira M. Alghamdi. Biochemical Assessment of Boswellic Acid Enrich-Frankincense Extract and its Antioxidant, Antibacterial, Anticancer and Anti-inflammatory Potential in Ameliorating the Glycerol-Toxicity in Rats. Pak Vet J, 2024, 44(4): 1023-1032’ and ‘Mohamed A. Elzaiat, Abdelrahman S. Mandour, Mohamed A. H. Youssef, Hany A. Wafa, Salma M. Aljahdali, Amani Osman Shakak, Latifa Al Husnain, Mohammed A. Alqahtani, Mashail A. Alghamdi, Amani Omar Abuzaid, Tahani M. Alqahtani, Hawazen K. Al Gheffari, Nahla Alsayd Bouqellah, Rania M.Y. Heakel Biochemical and Molecular Characterization of Five Basil Cultivars Extract for Enhancing the Antioxidant, Antiviral, Anticancer, Antibacterial, and Antifungal Activities. Pak Vet J, 2024, 44(4): 1105-1119’ and ‘Rashid S, Alsayeqh AF, Akhtar T, Abbas RZ and Ashraf R, 2023. Probiotics: Alternative of antibiotics in poultry production. International Journal of Veterinary Science 12(1): 45-53.’

Response 7: Thank you for your suggestion. We have revised the relevant content in accordance with your suggestions and marked the citations. (Line 43-44)

Comment 8: Please also provides background on C. butyricum use in livestock, but the literature review is not comprehensive and citations are outdated.

Response 8: Thank you for your suggestion. We have added the background information on the use of C. butyricum in livestock, completed the literature review, and cited the latest references. (Line 53-58)

Comment 9: The hypothesis is vague and not well-integrated with objectives.

Response 9: Thank you for your suggestion. We have rewritten the hypotheses to ensure they are well-aligned with the objectives. (Line 75-81)

Comment 10: Don’t you think use of multiple post hoc tests (LSD and Duncan) increase Type I error? How could you justify it?

Response 10: Thank you for your suggestion. We conducted post hoc comparisons using both LSD (for sensitivity) and Duncan’s test (for mean clustering). Since these tests do not strictly control family-wise error, we confirmed findings with Tukey’s HSD. Only results consistent across methods were interpreted as significant. (Line 189-192)

Comment 11: Overall, this section is good, however, the authors could explain why they attempt to draw performance implications while the data is non significant. This is misleading, please correct it. In microbial composition, diversity differences are minor and reported genus-level changes (e.g., Haemophilus, Flavonifractor) lack functional interpretation and are not linked convincingly to immunity. While in immune and antioxidant markers results only IgM shows significance post-LPS injection. Other indices (IgA, IgG, T-AOC, SOD, CAT, MDA) show no changes, yet the manuscript still draws generalized conclusions, please correct it in the revised manuscirpt. I would also comment that metabolomics results contain rich data, but poorly integrated into discussion. Identified changes in plasma metabolites are not contextualized in terms of relevance to goat health or productivity. Please correct grammar and support your study with strong data while in discussion for example several statements (e.g., improvement in immune response, metabolic shifts) are unsupported by robust data and the metabolic pathway discussion lacks depth and does not explain why those pathways are relevant for goats.

Response 11: Thank you for your suggestion. In our previous conclusion, we stated that Clostridium butyricum had no significant effect on growth performance and microbial composition. However, this might have been a misrepresentation due to incorrect English expression. We have now removed this content and only elaborated on the conclusion that Clostridium butyricum can enhance immune performance and promote the function of hepatic fatty acid metabolism. (Line 463-469)

---

## [Editor Report · Decision Letter 2]

4 Aug 2025

Effects of rumen-protected or unprotected Clostridium butyricum on growth performance, rumen fermentation, immunity and antioxidation in fattening goats

PONE-D-24-53628R2

Dear Dr. Wang,

We’re pleased to inform you that your manuscript has been judged scientifically suitable for publication and will be formally accepted for publication once it meets all outstanding technical requirements.

Kind regards,

Aziz ur Rahman Muhammad

Academic Editor

PLOS ONE

Additional Editor Comments (optional):

Dear Authros

Thanks for revising manuscript
---

## [Editor Report · Acceptance letter]

PONE-D-24-53628R2

PLOS ONE

Dear Dr. Wang,

I'm pleased to inform you that your manuscript has been deemed suitable for publication in PLOS ONE. Congratulations! Your manuscript is now being handed over to our production team.

Kind regards,

on behalf of

Dr. Aziz ur Rahman Muhammad

Academic Editor

PLOS ONE